# MAGinator enables accurate profiling of de novo MAGs with strain-level phylogenies

Trine Zachariasen [1] ✉, Jakob Russel[2], Charisse Petersen [3], Gisle A. Vestergaard [1], Shiraz Shah [4], Pablo Atienza Lopez [5,6], Moschoula Passali [5], Stuart E. Turvey[3], Søren J. Sørensen [2], Ole Lund[1], Jakob Stokholm[2,4], Asker Brejnrod [1] & Jonathan Thorsen [4]

Metagenomic sequencing has provided great advantages in the characterisation of microbiomes, but currently available analysis tools lack the ability to combine subspecies-level taxonomic resolution and accurate abundance estimation with functional profiling of assembled genomes. To define the microbiome and its associations with human health, improved tools are needed to enable comprehensive understanding of the microbial composition and elucidation of the phylogenetic and functional relationships between the microbes. Here, we present MAGinator, a freely available tool, tailored for profiling of shotgun metagenomics datasets. MAGinator provides de novo identification of subspecies-level microbes and accurate abundance estimates of metagenome-assembled genomes (MAGs). MAGinator utilises the information from both gene- and contig-based methods yielding insight into both taxonomic profiles and the origin of genes and genetic content, used for inference of functional content of each sample by host organism. Additionally, MAGinator facilitates the reconstruction of phylogenetic relationships between the MAGs, providing a framework to identify clade-level differences.

DNA sequencing has revolutionised our ability to gain insight into microbial compositions without relying on the ability to cultivate organisms. To explore these compositions, various methods have been developed that either rely on databases of marker genes of known organisms or attempt to reconstruct the chromosomes directly from the short reads by first assembling them into longer contigs and then binning these based on co-occurrences or DNA composition. Mapping reads against marker gene databases with tools such as MetaPhlAn[1], MetaPhyler[2] and mOTUs[3] is a fast and effective way of recovering the microbial composition both because the library depth required can be quite shallow and because the computational requirements are smaller. However, such methodologies have limitations originating from the reliance on predefined databases, limited

ability to estimate abundances at higher taxonomic resolution[4,5], and the lack of information on the functional repertoire of the identified taxa. Conversely, de novo binning strategies require high sequencing depth but can recover high-quality metagenome-assembled genomes (MAGs) from which the functional gene content can be directly linked to a specific organism. Ideally, this can recover genomes at the subspecies level that can be used in downstream analysis to generate more specific hypotheses about associations with outcomes. One example of this is to be able to identify organisms, which have the capacity of degrading Human Milk Oligosaccharides (HMOs), which are an important energy source for breastfed infants. Especially *Bifidobacteria* have this functionality, where certain strains or subspecies have specific preferences for certain HMO types[6–9]. Previously, it has

[1]Department of Health and Technology, Section of Bioinformatics, Technical University of Denmark, Lyngby, Denmark. [2]Department of Biology, Section of Microbiology, University of Copenhagen, Copenhagen, Denmark. [3]Department of Pediatrics, BC Children's Hospital, University of British Columbia, 950 West 28th Avenue, Vancouver, BC, Canada. [4]COPSAC, Copenhagen Prospective Studies on Asthma in Childhood, Herlev and Gentofte Hospital, University of Copenhagen, Copenhagen, Denmark. [5]Danish Multiple Sclerosis Center, Department of Neurology, Copenhagen University Hospital, Rigshospitalet-Glostrup, Glostrup, Denmark. [6]Department of Food Science, University of Copenhagen, Copenhagen, Denmark. ✉e-mail: trine_zachariasen@hotmail.com

been established that the presence of *Bifidobacterium longum* sub-species *infantis* (*B. infantis*) together with breastfeeding, plays a crucial role in providing a protective effect to mitigate the impact of anti-biotics on the early-life gut microbiome[7]. This underlines the significance of being able to accurately profile the microbiome at higher resolutions than the species level.

In this work, we have developed a pipeline that takes MAGs and original reads as input and generates output including accurate abundance estimates, subspecies-level phylogenies and gene synteny clusters that can improve insights into the microbiome composition (Fig. 1 A–F). As MAGinator is dependent on the quality of the contig assembly and MAGs, the resolution and granularity of the results are influenced by these. We do this by grouping MAGs into clusters that are phylogenetically separated at a higher resolution than species and estimating the abundances of these. This is done by identifying a set of signature genes directly from the given data and refining them according to statistical modelling to pick the ideal set suitable for abundance estimation. The fidelity of our estimated abundances is demonstrated on the Critical Assessment of Metagenome Interpretation (CAMI) strain-madness dataset, where we benchmark MAGinator against similar tools. Additionally, we show the functionality of MAGinator on a public dataset of inflammatory bowel disease (IBD) patients, where we identify differentially abundant taxa between patients and controls at high phylogenetic resolution.

MAGinator also enables the creation of Single Nucleotide Variant (SNV's) resolution phylogenetic trees from the signature genes. They are used for additional stratification of the MAGs and can be associated with metadata to obtain subspecies-level differences. We exhibit MAGinator's ability to obtain subspecies-level resolutions for *Bifidobacterium* from two real-world infant datasets. In this case, the signature genes were found de novo for one dataset and were then utilised to obtain subspecies-level resolution in the other cohort.

By combining the information from both contigs and gene content we identify synteny clusters of genes within subspecies, yielding information on shared pathways for the genes. Additionally, we show how we can associate the functional content to the identified clades, to improve hypotheses-generation on the impact of organisms, illustrated using the COPSAC$_{2010}$ cohort.

## Results

### MAGinator can accurately detect strains in simulated data

The performance of MAGinator was evaluated against the top 10 taxonomic profilers found in the second round of CAMI[5] challenges using the simulated short-read 'strain-madness' dataset. This dataset has been selected as it represents a heterogeneous strain environment, making strain and species detection highly relevant.

Running the MAGinator pipeline on the strain-madness data, 73 MAG clusters were identified, of these 22 clusters were present with less than 3 reads in 3 samples, so the abundance was set to 0. Of these 51 remaining entities, 30 were assigned with strain-level annotation by CAMITAX[10].

The profilers were compared with OPAL[11] (Fig. 2). For the majority of the tools, the performance decreased as the taxonomic categories became less inclusive (Fig. 2B & Suppl. Figure 2). The L1 norm measures the total error from the predicted and true abundance at each rank. From genus to species level, we observed drops in the average completeness 82.7–45.6% and the average purity 73.6–36.5%. MAGinator had the best average completeness at genus (99.8%) and species levels (89.6%) (Suppl. Table 3). At the genus level, MAGinator ranked number 5 for purity at 80.1% and the best-performing tool for the species level at 90.1%. The LSHVec gsa[12] had the best performance for purity at the genus level with 100%; however, at species level it has a purity of 37.5%, ranking number 5 in this group (Supplementary Table 4).

### MAGinator improves detection of differentially abundant organisms

To demonstrate the advantages of quantifying bacterial taxa at high resolutions we have re-analysed a well-designed metagenomics study from Franzosa et al.[13]. We chose this because it has deep sequencing well-suited for de novo MAG construction and a discovery/replication design with two distinct cohorts. In the absence of ground truth, replicating discoveries is a compelling strategy for making sure that findings are not false discoveries.

Beta diversity analysis of the two abundance matrices (MAGinator vs. their matrix created using MetaPhlAn2[14]) revealed a similar separation for IBD patients vs healthy controls. For this study MAGinator produces abundance matrices of much higher dimensionality (2140 vs 201 taxa) because of the higher resolution in taxa identifications, therefore prevalence and/or abundance filtering might be relevant in MAGinator produced tables for noise reduction (Fig. 3A–C).

To illustrate the improved ability of MAGinator to identify differentially abundant taxa we performed a regular differential abundance (DA) hypothesis test with Wilcoxon's rank-sum test (Fig. 3D–F). We looked for differentially abundant taxa defined as significant in the discovery cohort and replicated in the independent validation cohort. In the original analysis, 18 taxa were successfully validated in the independent cohort. With MAGinator, this increased to 213 taxa (Fig. 3 D–F).

### MAGinator enables tracking of subspecies across datasets

*B. infantis* is a gut microbe particularly adapted to the infant's gut due to its ability to metabolise HMOs, which are complex sugars that infants cannot metabolise themselves[15,16]. These capabilities are different from other major subspecies including *B. longum*. To demonstrate the utility of subspecies abundance estimation in MAGinator, we identified the signature gene set from one deeply sequenced infant cohort (COPSAC$_{2010}$) and used it to track subspecies abundances on another infant cohort (CHILD) with shallower sequencing but more samples. In the MAGinator pipeline, we identified two MAG clusters; one annotated as *B. infantis* and one as *B. longum* with GTDB-tk. In MetaPhlAn output we identified only one overall abundance for the species *Bifidobacterium longum*. Correlation analysis of these

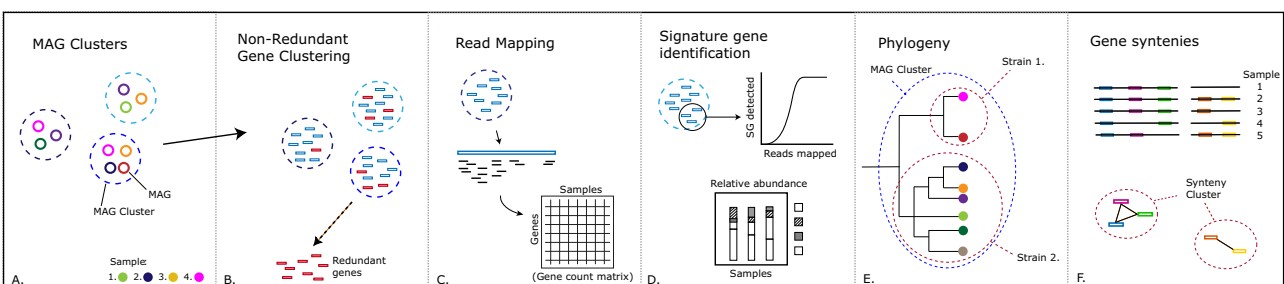

**Fig. 1 | Schematic visualisation of the main functions of the MAGinator workflow.** Following the data from (**A**) MAG clusters through (**B**) gene identification and clustering, **C** readmapping and counting, **D** Signature gene identification and abundance estimations, **E** Phylogenetic analysis and (**F**) Synteny cluster identification.

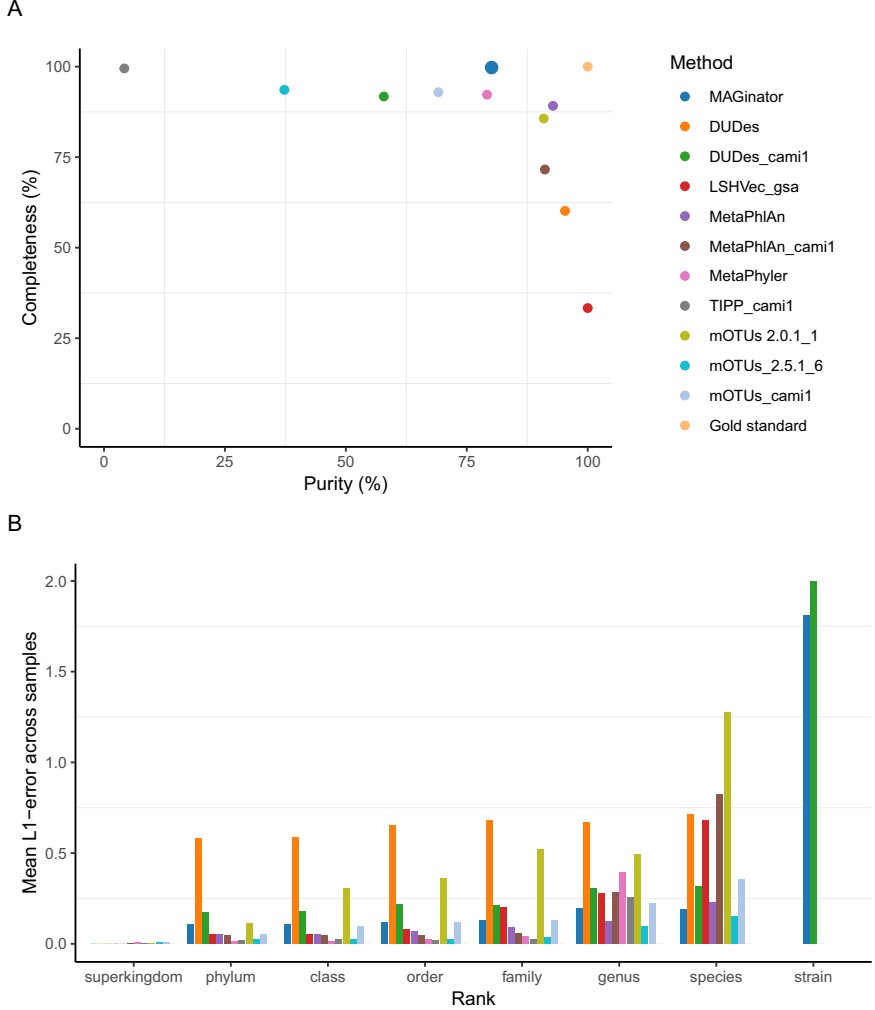

**Fig. 2 | Benchmark using OPAL for comparing taxonomic profiling results for the CAMI strain-madness data set. A** Purity and completeness of the profiles are shown at species-level (**B**) Mean of L1 norm error across samples for all ranks (0 indicating perfect reconstruction, 2 indicating totally incorrect reconstruction). Source data are provided as a Source Data file.

abundances shows that summed abundances of the two subspecies *B. infantis* and *B. longum* MAG clusters, explain 87% of the variance in the MetaPhlAn species (Supplementary Figure 2). In addition, we analysed the samples from both cohorts with StrainPhlAn[16] which detects strains in samples using prespecified species-level marker genes. Here, clustering of the sample-wise consensus sequences of the *B. longum* marker genes identified two clusters, one which clustered with reference strains of *B. longum* and one which clustered with reference strains of *B. infantis*. This result was previously shown for the CHILD cohort[7] and here we found similar results for COPSAC$_{2010}$ (Supplementary Figure 4). We hypothesised that this apparent duality represents the underlying balance of these two subspecies in each sample. We confirmed this by comparing the StrainPhlAn-clusters with the MAGinator relative abundances of all *Bifidobacterium* species, where we saw that the StrainPhlAn clusters depended on the ratio of *B. infantis* to *B. longum* (Fig. 4), but that more detailed information was accessible using the MAGinator derived relative abundances of each subspecies. This is an example of how de novo identification of subspecies-level MAG clusters and subsequent refinement of signature genes allows a higher resolution depiction of taxa for which the sequence coverage is sufficient in a subset of samples.

Additionally, we used the signature genes identified from the COPSAC cohort to track the two subspecies in the CHILD cohort. The relative abundances of the MAGinator clusters and the StrainPhlAn clusters were likewise examined (Suppl. Figure 4). When using the signature genes as a reference for the CHILD cohort MAGinator was still able to resolve the two subspecies into more well-defined clusters, yielding detailed profiling of the samples.

To estimate the fit of the signature genes for the two cohorts, we compared the read mappings and the presence of signature genes (Suppl. Figure 6A). The expected number of detected signature genes within a sample can be calculated from the number of reads that map to those genes using a negative binomial distribution[17]. We find that the COPSAC$_{2010}$ cohort deviates with a mean squared error (MSE) of 103.95, whereas the CHILD cohort deviates with a MSE of 878.09, indicating that the signature genes are better suited for profiling the specific subspecies found in the COPSAC cohort. To examine the cause of this large deviation for CHILD we created a heatmap of the read mappings to the signature genes (Suppl. Figure 6B). In accordance with Suppl. Figure 6A the samples cluster into two groups, which could be due to subspecies differences. Additionally, the genes are seen to cluster into multiple groups, where a group is seen to be absent in a large proportion of the samples, indicating that these genes have not been adequately selected for this subspecies for this dataset. Thus, mapping reads from a new data set onto signature genes from a previous data set can be an advantage when sequencing depth in the latter is too limited for good assembly. Reusing signature genes is also advantageous for easy comparison of abundances between data sets.

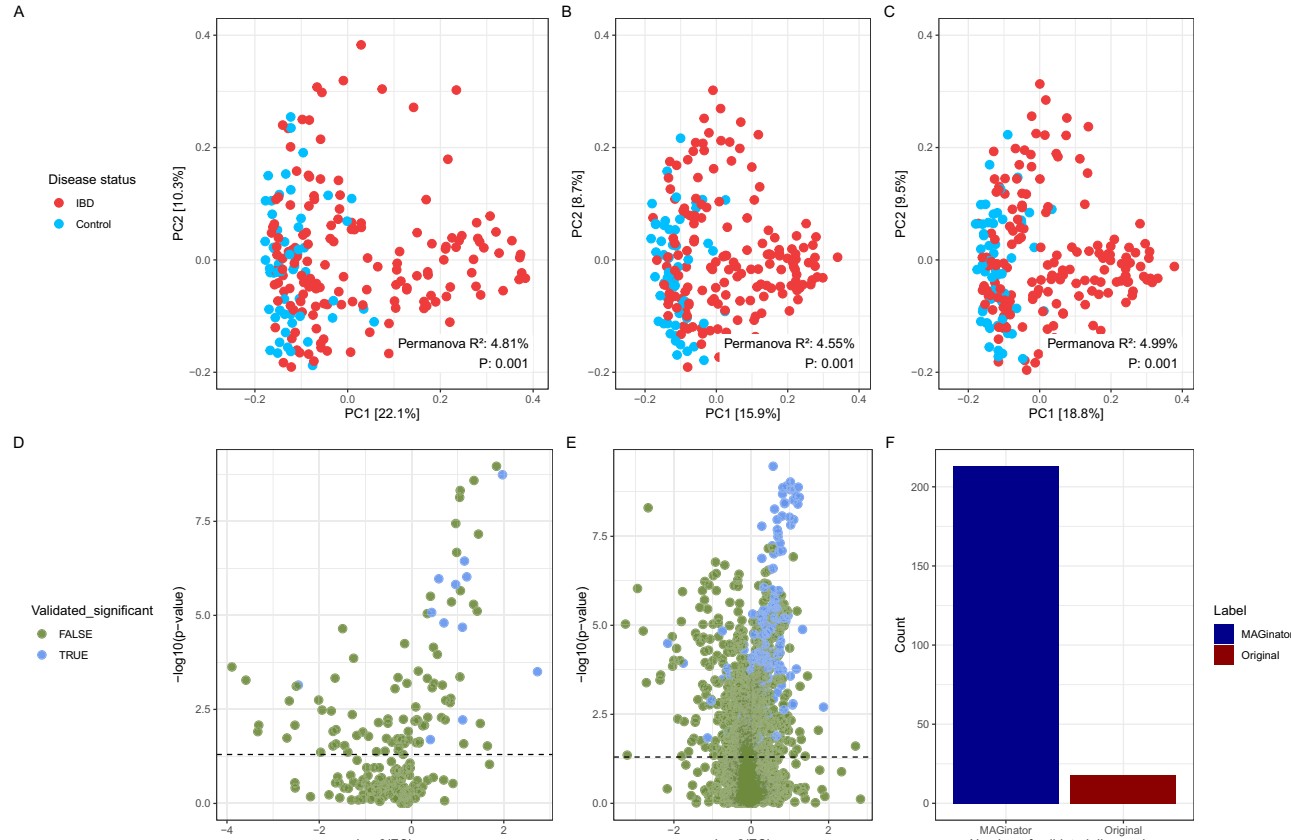

**Fig. 3 | IBD case study shows similar performance of MAGinator with beta diversity and improvements in DA analysis.** PCoA and PERMANOVA (999 permutations) for beta diversity analysis and Wilcoxon's rank-sum test for differential abundance analysis, where corrected $p$-values < 0.05 are considered significant. **A** PCoA of the original Franzosa et al. data (PERMANOVA, Jensen-Shannon distance, $F = 10.82$, $R^2 = 4.81\%$, $p = 0.001$) (**B**) MAGinator abundances (PERMANOVA, Jensen-Shannon distance, $F = 10.21$, $R^2 = 4.55\%$, $p = 0.001$) (**C**) filtered MAGinator abundances showing similar separation of IBD and control samples (PERMANOVA, Jensen-Shannon distance, $F = 11.23$, $R^2 = 4.99\%$, $p = 0.001$) (**D**) DA analysis of Franzosa et al. data, green points are taxa not significant in both cohorts (**E**) similar analysis on MAGinator abundances (**F**) Summary of validated discoveries using the two methods. Source data are provided as a Source Data file.

But optimal selection of adequately representative signature genes for a new data set requires running MAGinator de novo, if sequencing depth is adequate. Ideally, one might pool multiple data sets prior to running MAGinator in order to find signature genes that are equally representative for both data sets, making both abundance estimations and taxonomic entities directly comparable.

## MAGinator enables de novo discovery of strains from MAG cluster phylogenies

In the above case, MAGinator's ability to distinguish between subspecies depended on the binner to cluster the subspecies into two separate MAG clusters. This possibility may not be the case for other bacterial taxa and will vary between datasets. As an alternative, MAGinator provides samplewise phylogenies for each MAG cluster, where strains within the MAG cluster can be distinguished between samples. The result is presented as a maximum-likelihood tree and is based on sample SNVs within the signature genes, where each leaf corresponds to a sample. Because the analysis is based on read-mappings rather than assembled contigs, reliable phylogenies can be constructed even for samples where the taxon was not abundant enough to yield a MAG. E.g. for the COPSAC$_{2010}$ data set, the *Faecalibacterium* sp900758465 MAG cluster was found by VAMB[18] in 85 samples, but phylogenies were constructed for additional 13 samples (Suppl. Figure 7).

For the per-sample phylogenies to be reliable, signature genes must have adequate read coverage and sequencing depth. Also, samples must not contain mixtures of subspecies belonging to the same

MAG cluster. Thus cutoffs are set on alignment and SNV statistics to ensure reliability and can also be visualised alongside the tree as shown in Suppl. Figure 7 for visual confirmation. Using the shown median frequency of mixed SNVs it is possible to identify samples in which multiple variants of the MAG cluster were found. In COPSAC$_{2010}$, within-sample mixtures of MAG cluster variants were rare, yielding reliable sub-species-level information for most samples. Overall, for the 716 MAG clusters in the COPSAC data set, 387 MAG clusters had no samples containing mixed alleles. 329 MAG clusters harbour samples with mixed alleles, and within these particular MAG clusters an average of 38% of the samples had mixed alleles. In summary, across all MAG clusters, 4154 of 20,765 MAGs had mixed alleles (Suppl. Table 5).

### Strain diversity across environments

Within the 54 samples obtained from the honey-bee gut environment MAGinator identified 195 MAG clusters, in which 168 were found to have one or more samples with mixed alleles. For these MAG clusters, an average of 70% of the MAGs were found to have mixed alleles. For the 148 samples from the Tara Oceans expeditions 791 MAG clusters were found, from which 540 had at least one sample with mixed alleles. In total 37% of the MAGs in these clusters were found to contain mixed alleles (Suppl. Table 5).

### MAGinator identifies de novo gene synteny clusters aiding functional studies

MAGinator's signature gene identification step involves clustering all genes into clusters of conserved proteins. Such protein clusters are

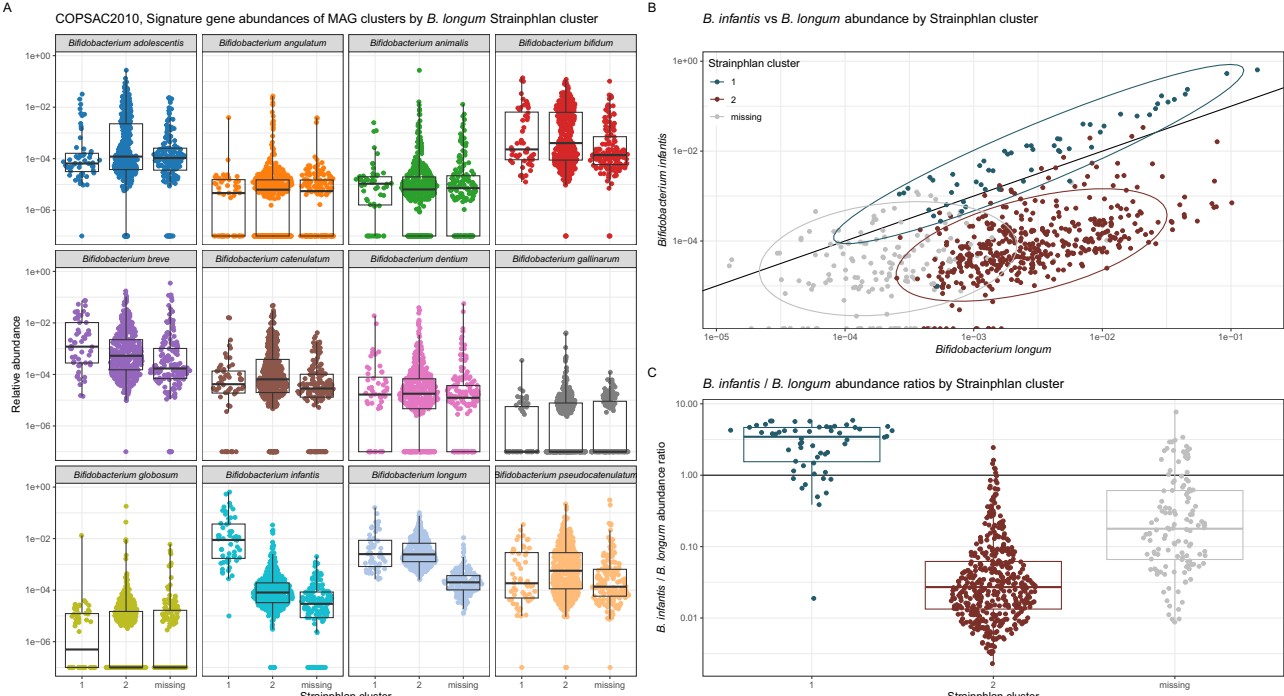

**Fig. 4 | Stratification of StrainPhlAn clusters using the relative abundances of Bifidobacterium longum subspecies from MAGinator Cluster 1 indicates B. infantis and Cluster 2 indicates B. longum.** Each point corresponds to a sample ($n = 662$). **A** Relative abundance of StrainPhlAn clusters stratified by all Bifidobacterium MAG clusters identified by MAGinator. Box plots indicate median (middle line), 25th, 75th percentile (box), and 1.5× interquartile range (whiskers) (**B**) Relative abundance of B. infantis and B. longum identified with MAGinator coloured by StrainPhlAn cluster. **C** The ratio of B. infantis to B. longum is displayed for the StrainPhlAn clusters. Source data are provided as a Source Data file.

orthologous (i.e. functionally conserved across different taxa) owing to conservative (and customisable) clustering parameters that maintain protein domain topology. Importantly, MAGinator's protein clusters are identified de novo. This means they include the protein "dark matter" ignored by traditional database-driven profiling, even though it comprises the majority of protein diversity in most metagenomic datasets to date. A key advantage of MAGinator's gene profiling is that each protein cluster, by definition, can be linked to the host MAG that encodes it. This enables the discovery of protein-host interactions against sample phenotypes, bridging taxonomic and functional profiling.

Genes can further be grouped into synteny clusters based on their genomic adjacency. Genes close to each other in the genome will be grouped into a synteny cluster, and they are usually part of the same pathway or have a related function. Part of the MAGinator workflow creates these synteny clusters. For the COPSAC[2010] cohort 746,251 synteny clusters were identified with an average of 3 genes per cluster (Supplementary Figure 8A, B). In order to evaluate the accuracy of the synteny clusters, functional gene annotations were performed using eggNOG[19] mapper. Subsequently, the predominant KEGG[20] module within each synteny cluster was determined, and the proportion of genes sharing the same annotation within the cluster was calculated (see Supplementary Figure 8C). Only synteny clusters with 5 or more genes and at least two annotated genes were included, qualifying 35,798 clusters for the analysis. For 28,341 clusters all genes in the synteny cluster were assigned the same KEGG module, and 80.5% of the modules had more than 80% agreement.

## Discussion
MAGinator is a pipeline for quantifying the abundances of de novo-generated MAG clusters. In contrast to reference-based abundance estimations, this allows extensive integration of abundance and functional properties for individual members of the microbial community.

Furthermore, it features the generation of signature gene-derived phylogenies for MAG clusters and the discovery of gene synteny clusters. It is implemented in Snakemake to take advantage of the integrated work distribution capabilities necessary for processing large-scale metagenomics data. It features logging for ease of monitoring progress and visualisation for diagnostic purposes. We have demonstrated the functionality and utility of MAGinator via several avenues, both simulated and real datasets.

The performance of MAGinator was evaluated in comparison to existing profiling tools. We benchmarked MAGinator using the simulated strain-madness dataset produced by CAMI II. We found that MAGinator is capable of profiling samples at a comparable level to the already established tools. Notably, while many tools performed well at the genus level, a decline in performance was observed when focusing on the species-level classification. This drop in performance is expected from reference-based methods, as they are limited to identifying only what already exists in their database and are thus unable to annotate novel species. MAGinator demonstrated a notable advantage in this regard, exhibiting the highest average completeness and purity when classifying samples at the species level. This indicates that MAGinator has the ability to achieve a more accurate and precise characterisation of microbial species present in the samples. It should be noted that the high completeness by MAGinator implies a greater sensitivity in detecting and including less abundant or rare taxa in the analysis. However, it may also introduce a certain level of noise or misclassification, which influences the estimation of beta diversity.

When examining the performance of MAGinator on a real dataset, the beta diversity was comparable to the analysis carried out by Franzosa et al. Reanalysing their data demonstrates how MAGinator can be used for a metagenomic association study. With the higher resolution of MAGinator when quantifying MAG clusters investigators have the possibility of discovering differentially abundant taxa in much richer detail without compromising other parts of a traditional analysis

such as PCoA. Depending on the intention of the study, and the taxonomic composition of the studied microbiomes, the high resolution can also be utilised to gain deeper insights into the subspecies taxonomies. This is for instance, relevant when analysing the *Bifidobacterium longum* subspecies.

*B. infantis* is highly relevant to investigate, as it is known for its greater capacity to metabolise HMOs compared with its closely related subspecies, such as *B. longum*. As their genomes are very similar, distinguishing them by database-dependent approaches is challenging. With StrainPhlAn, we are able to identify 2 mutually exclusive clusters, each representing a subspecies. However, we see that the two MAG clusters identified with MAGinator for B. infantis and B. longum yield higher resolution in the form of individual abundance estimates for each. MAGinator is able to successfully classify samples containing the subspecies in samples with low abundance and even when a MAG is not produced in that sample.

These results were reproduced in the CHILD cohort using the signature genes identified in COPSAC$_{2010}$ for the two subspecies. As samples from the CHILD cohort used in this study had lower sequencing depth, still being able to separate the subspecies is valuable. Importantly, it is worth noticing that the separation would most likely have been stronger if the signature genes had been found de novo for the specific cohort. This is supported by the read mappings to the signature genes showing a subset of the signature genes defined in COPSAC$_{2010}$ missing in the CHILD cohort, which presumably resulted in an underestimation of the abundance for a subset of the samples. This phenomenon highlights the importance of de novo dataset-specific discovery of signature genes to yield the best possible abundance estimates of closely related taxonomic entities. A similar phenomenon would be expected when using database-derived strain marker genes.

From the COPSAC$_{2010}$ cohort we demonstrated MAGinator's ability to create SNV-level trees based on the sequences from the signature genes of a MAG cluster, used for more fine-grained stratification of the MAGs. Even in samples where no MAG is assembled, a reliable phylogeny can still be derived when enough reads map to the signature genes. By placing these samples in the tree, information from the closely related MAGs can be utilised to find strain-level entities, even for low-abundance samples. Tree distances can be tested against sample meta-data using e.g. PERMANOVA, thus revealing whether the added subspecies resolution is informative for the research question at hand. If so, cutting the tree at evolutionarily sensible depths could define subspecies or strains de novo that drive specific sample phenotypes. Coupling this information with MAG gene content allows for the discovery of clade-specific genes, enabling their identification in new data sets.

From the alignment of the signature genes it is also possible to identify the extent of strain-diversity within the MAG cluster, by identifying samples which display a certain frequency of mixed alleles. Samples with mixed alleles harbours multiple strains of the MAG cluster.

For the COPSAC$_{2010}$ dataset, we found that within-sample strain diversity was low. When comparing allele frequencies across other environments, like the honey-bee gut and the ocean a greater strain diversity is seen. While the honey-bee gut exhibits the highest proportion of MAG clusters with mixed alleles (86%), the Tara Oceans exhibit a higher proportion (68%) compared to that of COPSAC$_{2010}$ (46%). Notably, the percentage of samples in MAG clusters containing MAGs that had mixed alleles from Tara Oceans and COPSAC2010 was comparable, with 37% and 38%, respectively, whereas that number for honey-bees was 71%. These findings underline the effect of selective pressure within different environments on both strain- and species-level diversity within the microbiomes.

Additionally, the COPSAC$_{2010}$ cohort was used to illustrate MAGinator's ability to group genes co-localised on the chromosome

into synteny clusters, further combining the strengths of using both genes and contigs. As genes found close together are often part of the same genetic pathway or share the same function, this is a valuable insight for associating organisms with the outcomes of a study. This has been validated by functionally annotating the genes of the predicted synteny clusters, confirming that the genes found in synteny are often annotated to be part of the same metabolic pathway. Currently, accuracy is limited by MAGinator's lack of operon awareness. As bacterial operon prediction methods improve, these could be integrated into MAGinator and eliminate such noise. Users can also eliminate noise at the expense of sensitivity by altering a number of user-modifiable parameters. Crucially, MAGinator's protein and synteny clusters are de novo, meaning that they do not need to yield any known database hits, as is often the case for new virulence factors or antiviral defence systems. Users may find that such "dark matter" gene clusters yield particularly strong associations against sample meta-data, making them prime candidates for downstream genetic or biochemical studies aimed at deciphering their mechanisms of action.

In conclusion, we have described the development of MAGinator −a pipeline for quantifying MAG clusters and demonstrated the benefits of this approach to commonly generated data types in the metagenomics field. Through reanalysis of publicly available data, we have illustrated how insights can be gained from MAGinator at a higher taxonomic resolution than available from commonly used tools. We believe that this higher resolution is key to unlocking the potential of metagenomics to identify critical subspecies for human health and environmental investigations. MAG cluster resolution metagenomics allows for accurate integration of abundance, taxonomic and functional annotation in microbiome studies, which is needed to empower investigations in the microbiome field.

## Methods

### Implementation

***Input.*** The input to the MAGinator workflow comprises a set of samples with (1) shotgun metagenomic sequenced reads, (2) their sample-wise assembled contigs, and (3) sample-wise MAGs (groups of contigs from the same genome), clustered across samples, as defined by a metagenomic binning tool (see below).

Reads should be provided in a comma-separated file giving the location of the fastq files and formatted as: SampleName,PathTo-ForwardReads,PathToReverseReads. The contigs should be nucleotide sequences in FASTA format. The MAGs should be given as a tab-separated file including the MAG identifier and contig identifier. The sample-wise MAGs should be grouped into MAG clusters representing a taxonomic entity found across the samples, which will usually be species but can also be at the subspecies level, depending on the characteristics of the input data. MAGinator is flexible regarding which tool is being used for creating the MAGs, however we recommend using VAMB[18]. If other binners are used, MAG clustering across samples would have to be implemented before running VAMB. As MAGinator relies on the input MAGs a larger sample size is recommended. The specific number of samples relies both on the sequencing depth and the diversity of the community being analysed. We advise the user to look at the number of MAG clusters created and assess them according to the environment being analysed.

***Dependencies.*** The dependencies to run MAGinator are mamba[21] and Snakemake[22]−all other dependencies are installed automatically by Snakemake through MAGinator. Additionally, MAGinator needs the GTDB-tk database downloaded for taxonomic annotation of MAGs and as a reference for the phylogenetic SNV-level analysis of the signature genes.

***Output generated.*** MAGinator generates multiple outputs and intermediate files useful for additional downstream analysis

(Supplementary Table 1, Supplementary Figure 1). Importantly, MAGinator outputs the taxonomy of the MAGs, the signature genes of the MAG clusters, the sample-wise relative abundances of the MAG clusters, a non-redundant gene matrix with sample-wise mapping counts, synteny clusters and inferred phylogenies for each MAG cluster along with a table presenting samples showing evidence of strain mixtures within each MAG cluster. Additionally, a folder is created containing the log information of all the jobs run by Snakemake.

**Application**. MAGinator is written in Python 3. It is based on a set of Snakemake[22] workflows and is easily scalable to work for both single servers and compute clusters. MAGinator is implemented as a python package and is available on GitHub at https://github.com/Russel88/MAGinator. The user can adjust the individual steps of the pipeline using various parameters (Suppl. Table 2). The results in this paper are based on MAGinator v.0.1.10.

The MAGs are filtered based on a minimum size for inclusion, with a default size of 200,000 bp. The included MAGs are taxonomically annotated using GTDB-tk (v.2.1.1)[23], by calling genes using Prodigal (v.2.6.3)[24], identifying GTDB marker genes and placing them in a reference tree. As the taxonomic annotation of the MAG clusters is found to be redundant, clusters with the same taxonomic assignment can be combined into one cluster, with the flag '--mgs_collections' which we identify as a Metagenomic Species (MGS). Redundant genes are identified by clustering with MMSeqs2 (v.13.45111)[25] easy-linclust using a default clustering-coverage and sequence identity threshold of 0.8, creating a list of the representative genes along with their cluster-members. The redundant genes are filtered away, leaving a non-redundant gene catalogue. The raw reads are mapped to the gene catalogue using BWA mem2 (v.2.2.1)[26] and counted using Samtools (v.1.10)[27], leaving a gene count matrix, which is used as input for the signature gene refinement and following phylogenetic clade separation and abundance estimates.

### Signature gene identification

We previously described the method for identifying the signature genes for the data set[17]. In brief, signature genes are selected to ensure that they 1) are unique for the MAG cluster, 2) are present in all members of the cluster, and 3) are single-copy.

To accomplish this, the following steps are taken: Initially, the non-redundant gene count matrix is curated to discard any genes if they have (redundant) cluster members originating from more than one MAG cluster, as they are thus not specific for that biological entity. Subsequently, the remaining genes within each MAG cluster are sorted based on their co-abundance correlation across the samples. As the genes are unique for the species, if they are consistently detected in similar abundance across samples, it suggests that they are single-copy. This step also mitigates differences in reading mappings caused by biological or technical variations. The initial set of signature genes for each biological entity is selected from the most correlated genes. Subsequently, these signature genes are further refined and optimised by fitting them to a rank-based negative binomial model that captures the characteristics of the specific microbial composition in the input data. The signature gene set is evaluated across the samples, by calculating the probability of the detected number of signature genes given the number of reads mapping to the MAG cluster. Finally the abundance of each MAG cluster is derived from the read counts to the identified signature genes normalised according to the gene lengths.

### SNV-level resolution phylogenetic trees

To elucidate the smaller biological differences within the MAG clusters, MAGinator will infer a phylogeny based on the sequences of the signature genes. Based on the read mappings to the signature genes the sample-specific SNVs are called using output from Samtools mpileup. An alignment for each signature gene is made for all samples containing the signature genes using MAFFT (v.7)[28] run with the offset value of 0.123 as no long indels are expected. MAGinator allows phylogenetic inference to be calculated with either the fast method Fast-Tree (v.2)[29] (default) or the more accurate but resource-intensive method IQ-TREE (v.2)[30]. In samples where no MAG was found, the phylogenies can be used to detect rare subspecies-level entities based on just a few reads mapping to the signature genes and to infer functions and genes from closely related MAGs from other samples. The criteria for inclusion in the tree can be adjusted by the user. For a sample to be included in the phylogeny the following three criteria have to be met 1) minimum fraction of non-N characters in the alignment, 2) minimum number of GTDB marker genes to be detected, 3) minimum number of signature genes to be detected. The default values for a sample to be included in the phylogenetic tree have been set relatively low in order to enable the placement of samples in the tree, even in cases of very low abundance. The trees can be associated with metadata to obtain clade-level differences associated with study design variables such as disease phenotype, sampling location, or environmental factors.

### Gene synteny

Based on the gene clustering with MMSeqs2 a weighted graph is created, which reflects the adjacency of the genes on contigs. If genes are close enough in the graph, they will be categorised as part of the same synteny cluster, and it is assumed that they have related functionality and/or are part of the same functional module. Clustering is determined using mcl (v.14)[31], where the user has the options to influence the adjacency count and stringency of the clusters. Only immediate adjacency is considered. By default, genes found adjacent just once are included in the graph, but this can be tuned to make more strict clusters. The inflation parameter for MCL-clustering of the synteny graph is important for the size of the gene clusters and is, by default, set high in order to yield small and consistent clusters.

### Taxonomic scope of gene clusters

The taxonomic assignment of the sample-specific MAG is done using GTDB-tk. In some cases it will not be possible to assign a taxonomy to the MAG, which could be due to contamination, the MAG originating from a currently undescribed organism or due to too little information found in the MAG. In these cases an alternative is to assign the gene clusters, found in the MAG, a taxonomy. The taxonomic scope of the genes is described for the category in which they are predominantly found in, given by a fraction defined by the user (default value 0.9). E.g. if run with default options and a gene cluster has the assignment "Bacteria Firmicutes_A Clostridia Lachnospirales Lachnospiraceae Anaerostipes NA", then at least 90% of the genes should be found in Anaerostipes. The algorithm will find the most specific taxonomic rank which has at least 90% agreement across the genes in the cluster assigned by GTDB-tk.

### Workflow design

The MAGinator workflow has been constructed to make the information flow between the different modules automatically (Suppl. Figure 1).

The data goes through a series of filtering and processing steps (Fig. 1 A−F), including:

A: MAG clusters, which are composed of one or more MAGs, are inputted.

B: The genes are clustered and redundant genes are removed.

C: Reads are mapped to the genes, creating a gene count matrix.

D: Signature genes are identified for each MAG cluster, and used for abundance estimations

E: Based on the signature genes, SNV-level resolution phylogenetic trees are created, and the taxonomic scope of gene clusters is identified.

F: Synteny-clusters of genes are identified, reflecting the adjacency of the genes on the contigs.

## Benchmarking on CAMI's simulated strain-madness data set

The construction of the strain-madness benchmarking dataset was part of the second round of CAMI challenges[5]. The data consists of 100 simulated metagenomics samples consisting of paired-end short reads of 150 bp. The samples were run through a preprocessing workflow prior to the analysis. This involved the removal of adaptors with BBDuk (v. 38.96 http://jgi.doe.gov/data-and-tools/bb-tools/) run with the following settings 'ktrim=r k = 23 mink=11 hdist=1 hdist2 = 0 ptpe tbo', removal of low-quality and short reads (<75 base pairs) with Sickle (v. 1.33)[32] and removal of human contamination (reference version: UCSC hg19, GRCh37.p13) using BBmap (http://jgi.doe.gov/data-and-tools/bb-tools/) leaving an average of 6.6 million reads (SD: ±2802 reads) per sample.

To generate de novo assemblies, Spades (v. 3.15.5)[33] was utilised with the -meta option, with kmer sizes of 21, 33, 55 and 77, and contigs shorter than 1500 bp being discarded. Read-to-assembly mapping was carried out using BWA-mem2 (v.2.2.1)[26] and SAMTOOLS (v.1.10)[27]. Contig depths were assessed using Metabat2's jgi_summarize_bam_contig_depths (v.2.12)[34], while contigs were binned into MAGs using VAMB (v.3.0.8)[18] with default settings.

The reads, contigs and MAGs were run through the MAGinator workflow (v.0.1.16). For comparison purposes, the VAMB clusters were annotated with an NCBI Taxonomy ID using CAMITAX[10]. The profile was created with an R custom script and the lineage was found using NCBI's taxonomy toolkit (https://bioinf.shenwei.me/taxonkit). As the strain identifiers from the gold standard do not exist in the NCBI database (e.g. 1313.1), we have assigned an extra number to the Taxonomy ID for the clusters which had the same species-level annotation, starting at 1 to the number of redundantly annotated clusters.

The data for the benchmarking was obtained from CAMI second challenge evaluation of profiles. The profiles used for the benchmarking in this study were selected based on the best-performing tools found in the CAMI II paper. The top 10 profiles comprise DUDes[35] (v.0.08), LSHVec[12], MetaPhlAn2[14] (v.2.9.22), MetaPhyler[2] (v.1.25), mOTUs[3] (v.2.0.1 and v.2.5.1) and TIPP[36] (v.4.3.10). The profiles were compared using Open-community Profiling Assessment tooL (OPAL) (v.1.0.11), which was run with default settings.

## Franzosa et al. reanalysis

Processed taxa and metadata tables were obtained from the Franzosa et al.[13]. supplementary materials. Raw data were downloaded from ENA using the provided accessions, and run through the preprocessing, assembly and binning before running the entire MAGinator pipeline. Four samples failed the assembly (PRISM|7238, PRISM|7445, PRISM|7947, PRISM|8550) and were excluded from all downstream analyses, both in the original and the MAGinator processed tables, leaving 216 samples.

## Statistical methods for abundance matrices

Abundance matrices were analysed in R (v.4.1.2). Sample management and beta diversity calculations were done in {phyloseq}[37], along with PCoA analysis. Differential abundance testing was done with the {DAtest} R package, which uses the Wilcoxon test function (Wilcox.test) from the {stats} package, with $p$-values adjusted by Benjamini-Hochberg false discovery rate correction. Corrected $p$-values less than 0.05 were considered significant.

## Subspecies resolution of Bifidobacterium longum

***COPSAC dataset - data characteristics and preparation***. The COPSAC$_{2010}$ cohort consists of 700 unselected children recruited during pregnancy week 24 and followed closely throughout childhood with extensive sample collection, exposure assessments and longitudinal clinical phenotyping[38–40]. From the cohort, we used 662 deeply sequenced metagenomics samples taken at 1 year of age. The details of the study and sequencing protocol have previously been published[40]. The samples consist of 150-bp paired-end reads per with mean ± SD: 48 ± 15.5 million reads.

The data was analysed using the same approach as for the strain-madness data set, with the exception of filtering away reads shorter than 50 bp in the preprocessing step. This workflow yielded 880 MAG clusters for the samples.

MAGinator was run using the reads, contigs and MAGs from VAMB as input. Thus creating a set of signature genes for each MAG cluster which has been found de novo for this particular dataset.

**CHILD dataset - data characteristics and preparation.** The Canadian Healthy Infant Longitudinal Development (CHILD) study comprises a large longitudinal birth cohort with stool collection in infancy for microbiome analysis[41]. Stool samples used in this analysis were sequenced to an average depth of 4.85 million reads (SD: 1.79 million), and samples which included >1 million reads after preprocessing were kept for the current analysis[7].

We analysed a subset of the CHILD cohort, consisting of 2846 metagenomic sequenced faecal samples from infants. To overcome the shallow sequencing, the signature genes of the COPSAC$_{2010}$ cohort were used to profile the samples instead of running MAGinator. To ensure that the process of the read mappings was identical to COPSAC, the read mapping was carried out using the full gene catalogue. Next, the read counts for the signature genes were extracted and used to derive sample-wise abundances for each MAG cluster.

**Examining bifidobacterium MAG clusters.** The detection of signature genes for *B. infantis* for the COPSAC$_{2010}$ (n = 662) and CHILD (n = 2846) cohorts was carried out by creating a binary detection matrix and using the standard function (heatmap) with default values in R. Furthermore, we compared the abundances of all the *Bifidobacterium* MAG clusters derived from MAGinator with abundance estimates from Metaphlan 3 (v.3.0.7) and subspecies phylogenies from Strainphlan 3 (v.3.0.7) for the species *Bifidobacterium longum*. The phylogenetic tree output by Strainphlan was converted into a distance matrix and clustered using partitioning around medoids into two clusters. The two clusters were annotated as *B. longum* subsp. *longum* (*B. longum*) and *B. infantis* based on the placement of *Bifidobacterium longum* reference genomes in the phylogenetic tree.

## SNV-level phylogenetic trees for COPSAC dataset

For each MAG cluster, the sequences of the signature genes were used as a reference to create an SNV-level phylogenetic tree. The trees for COPSAC$_{2010}$ were constructed with the default values of MAGinator, producing both a tree in Newick file format for each MAG cluster and files containing the statistics for the alignments. The tree for *Faecalibacterium* sp900758465 was visualised in R using {ggtree}[42]. The heatmaps in Suppl. Figure 7 was constructed from B) stats.tab and C) stats_genes.tab. The median frequency of bases in the signature gene alignment with mixed alleles was calculated based on positions with a depth of minimum 2 and normalised according to the gene length. A major allele frequency of at least 0.8 was required for the sample to be considered homogenous. These are also the default cutoffs, and users can adjust them to trade off sensitivity for specificity.

## Strain mixtures within de novo MAGs across environments

To assess the degree of within-MAG cluster strain diversity for non-human associated environments, two public datasets were included in the analysis. One study done by Engel and Ellegaard examined the honey-bee gut[43] and the Tara Oceans study[44]. The raw data was run

through the same workflow as the strain-madness data and run through MAGinator. Due to computational limitations and the size of the Tara Oceans samples only 148 of 243 samples were successfully assembled.

### Gene syntenies and functional annotation for COPSAC dataset

The non-redundant genes were annotated using eggNOG mapper (v.2.0.2)[19,45,46]. Of the 14.7 million non-redundant genes 9.2 million were annotated. The visualisation of the synteny clusters was done with {igraph}[47].

### Statistics and reproducibility

The statistical methods included in this study has been conducted with R (v.4.1.2). In this study we have analysed 5 public datasets, COPSAC$_{2010}$[38] ($n = 662$), CHILD[41] ($n = 2846$), Franzosa et al. IBD-study[13] ($n = 220$), Tara Oceans[44] ($n = 243$) and honey-bee[43] ($n = 54$). For Franzosa et al. and Tara Oceans not all samples succeeded in assembly and was thus not included in the analysis included in this study, leaving 216 and 148 samples respectively.

### Reporting summary

Further information on research design is available in the Nature Portfolio Reporting Summary linked to this article.

## Data availability

All relevant data supporting the key findings of this study are available within the article and its Supplementary Information files. Supplementary dataset 1 contain the Supplementary Figs. and tables. The CAMI II strain-madness benchmarking dataset is available at https://frl.publisso.de/data/frl:6425521/strain/short_read/. The gold standard and benchmark profiles are found at https://github.com/CAMI-challenge/second_challenge_evaluation/tree/master/profiling. The dataset from Franzosa et al. used for benchmarking is available as supplementary from their paper and the raw data is available at ENA accession SAMN08049618. The raw COPSAC fastq files are available at NCBI under BioProject PRJNA715601. The honey-bee data is publicly available and found in the sequence read archive (SRA) with the accession SRP150166. The Tara Oceans data set is publicly available and found at ENA with Study accession PRJEB1787. The CHILD shotgun metagenomics sequencing data is available at NCBI BioProject PRJNA838575 . Source data are provided in this paper. **Availability and implementation:** MAGinator is available as a Python module at https://github.com/Russel88/MAGinator.

## Code availability

MAGinator is available at GitHub (https://github.com/Russel88/MAGinator)[48].

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

## Acknowledgements

We express our deepest gratitude to the children and families of the COPSAC cohort studies for all their support and commitment. We acknowledge and appreciate the unique efforts of the COPSAC research team. All funding received by COPSAC is listed on www.copsac.com. The Lundbeck Foundation (Grant no R16-A1694); The Ministry of Health (Grant no 903516); Danish Council for Strategic Research (Grant no 0603-00280B) and The Capital Region Research Foundation have provided core support to the COPSAC research centre. JS has received funding from the Danish Council for Independent Research (Grant no. 8045-00081B). We thank the CHILD Cohort Study (CHILD) participant families for their dedication and commitment to advancing health research. CHILD was initially funded by CIHR and AllerGen NCE, and the metagenomic data reported here was generated with support from Genome Canada and Genome BC (274CHI).

## Author contributions

The figures and tables were created by T.Z, P.A.L, A.B and J.T. T.Z, A.B, J.R, J.T and S.S draughted the manuscript. The MAGinator software was developed and set up by T.Z and J.R. T.Z, J.R, C.P, G.V, S.S, P.A.L, M.P, S.T, S.J.S, O.L, J.S, A.B and J.T provided intellectual input and aided in the theoretical aspects of shaping this study. The corresponding author had full access to the data and held the final responsibility for deciding to submit the manuscript for publication. T.Z, J.R, C.P, G.V, S.S, P.A.L, M.P, S.T, S.J.S, O.L, J.S, A.B and J.T guarantee that the accuracy and integrity of any part of the work have been appropriately investigated and resolved and all have approved the final version of the manuscript. None of the authors received any honorarium, grant, or other forms of payment for creating this manuscript.

## Competing interests

The authors declare no competing interests.
