## [Peer Review File · Nature Communications]

MAGinator enables accurate profiling of de novo MAGs with strain-level phylogeniesReviewer #1 (Remarks to the Author):

In the manuscript, the authors described MAGinator, a bioinformatic pipeline for the profiling of composition of metagenomic datasets. Specifically, MAGinator takes a set of sample-wise MAGs and the corresponding short-reads as inputs. MAGinator then use a set of genes that are specific to the MAG clusters (defined by the user) to assess their abundance.

Based on my understanding, the authors assume that a MAG represent a group of contigs from the same genome (line 100). I was wondering if I could interpret "the same genome" as "the genome of a strain in a community (which is most likely not the case in reality)"? I'm concerned that the authors may have misunderstood the interpretation of MAGs obtained from metagenomic datasets. However, I'm open to any clarifications or corrections from the authors, and I apologize if I misunderstood how MAGinator works.

It is very often seen that, especially in the presence of closely related populations, contigs assembled from metagenomes are actually "chimeras" of multiple closely related populations (especially for short-read sequencing), to be specific, some bases in a contig are derived from one population while others are from a different closely related population.

If the input contigs (not even saying MAGs) into MAGinator are not corresponding to actual strains (or subpopulations) in a community, I was wondering what the "strain" means in the title and Figure 1E. We can surely see the existence of closely related (sub)population in a metagenome (by mapping reads back to the contigs), however, we might not be able to differentiates their genome content since their sequences were coupled together in the assembly and binning steps.

Given this, I was also wondering how the relative abundance of subpopulation could be calculated. Imagine a community comprising two closely related subpopulations, A and B, with relative abundances of 70% and 30% respectively. In the process of reads assembly, a contig was obtained where the 1000th base corresponds to subpopulation A and the 2000th base corresponds to subpopulation B. If MAGinator assumes that a contig is always derived from the same strain, will MAGinator report two populations in a community with equal abundance $((30 + 70)/2)$ (which is apparently not the case in reality)?

I want to once again highlight that I am receptive to any clarifications from the authors, and I sincerely hope that my understanding of how MAGinator works might be incorrect.

Best,
Reviewer

Reviewer #2 (Remarks to the Author):

The authors describe MAGinator, a new method for profiling of metagenomic data sets, able to identify strain-level variation and abundance estimates. The article is well-written, but could be a little more concise in parts, e.g. the introduction of HMO-associated microbes, some command line options (could be moved to the supplement) and the strain tracking results. The presented results are impressive and a significant improvement over the state-of-the-art for subspecies level profiling.

The software is available on GitHub and indeed installed easily. Unfortunately, I was not able to reproduce the results because of run time restrictions on the server available to me. Some information about the estimated running time should be included.

There are a few additional points which should be addressed:

Page 4: Has MAGinator been tested with other binners than VAMB? How does binning performance/MAG quality influence performance of the pipeline?

Page 5: The sentence "if they are consistently detected in similar abundance across samples, it suggests that they are single-copy" is not clear to me; a gene consistently occurring multiple times would also occur in similar abundances across samples?

Page 6: Why are no long indels expected between strains in different samples? The acquisition of e.g. *amr* genes in response to external influences seems possible.

The reasoning for the values why a sample is included into the phylogeny should be explained somewhere

Page 7: Presented are the results on the CAMI 2 strain madness data set, but not on the other CAMI 2 data sets with fewer samples. Since the other data sets also consist of many samples, I am wondering whether there is an effect of the number of samples on the results by MAGinator. Is there a "minimum" number of samples for which MAGinator produces good results?

Figure 2/Suppl. Table 2/3: DUDes_cami1 reports a L1-error in Figure 2B, but in Suppl. Table 2 it has 0% completeness and 0% purity, how is this possible? Is there a reason for the - relatively - poor performance in terms of L1-error of MAGinator for the high phylogenetic ranks?

For the new gene synteny clusters, 80% of modules having 80% agreement is a nice result, though there are still >7,000 clusters for which there is no corresponding KEGG module, so some way to estimate or reduce the number of false positives would help.

REVIEWER COMMENTS

Reviewer #1 (Remarks to the Author):

In the manuscript, the authors described MAGinator, a bioinformatic pipeline for the profiling of composition of metagenomic datasets. Specifically, MAGinator takes a set of sample-wise MAGs and the corresponding short-reads as inputs. MAGinator then use a set of genes that are specific to the MAG clusters (defined by the user) to assess their abundance.

Based on my understanding, the authors assume that a MAG represent a group of contigs from the same genome (line 100). I was wondering if I could interpret “the same genome” as “the genome of a strain in a community (which is most likely not the case in reality)”? I'm concerned that the authors may have misunderstood the interpretation of MAGs obtained from metagenomic datasets. However, I'm open to any clarifications or corrections from the authors, and I apologize if I misunderstood how MAGinator works.

It is very often seen that, especially in the presence of closely related populations, contigs assembled from metagenomes are actually “chimeras” of multiple closely related populations (especially for short-read sequencing), to be specific, some bases in a contig are derived from one population while others are from a different closely related population.

If the input contigs (not even saying MAGs) into MAGinator are not corresponding to actual strains (or subpopulations) in a community, I was wondering what the “strain” means in the title and Figure 1E. We can surely see the existence of closely related (sub)population in a metagenome (by mapping reads back to the contigs), however, we might not be able to differentiates their genome content since their sequences were coupled together in the assembly and binning steps.

Reply 1) In response to your valuable insights, we would like to emphasize that the efficacy of MAGinator is reliant upon the quality of the initial data it processes, including reads, contigs, and their assemblies.

Regarding your point about strains, we agree with you. The term 'strain' is hard to define and MAGinator might not adequately capture the nuanced diversity of a strain. Therefore, we have made the adjustment to refer to 'subspecies' to better align with the complex nature of these genomic variations.

Given this, I was also wondering how the relative abundance of subpopulation could be calculated. Imagine a community comprising two closely related subpopulations, A and B, with relative abundances of 70% and 30% respectively. In the process of reads assembly, a contig was obtained where the 1000th base corresponds to subpopulation A and the 2000th base corresponds to subpopulation B. If MAGinator assumes that a contig is always derived from the same strain, will MAGinator report two populations in a community with equal abundance $((30 + 70)/2)$ (which is apparently not the case in reality)?

Reply 2) In the hypothetical scenario described, MAGinator faces a limitation in distinguishing the abundances between closely related subpopulations within a single contig. This limitation arises from the fact that MAGinator operates at the resolution of the initial assembly or binning process. Even with reference-based methods that are the standard today (mOTUs, MetaPhlAn), differentiating between such subpopulations would be limited by the database and alignment criteria. Similarly, MAGinator cannot not accurately report the individual abundances of such subpopulations, as its resolution is constrained by the initial assembly, quality of binning and MAG clustering granularity. It is important to note that MAGinator's population granularity is inherited from VAMB, making it data-driven and thus highly meaningful for the ecological niche studied. In addition, owing to the phylogenetic profiling provided by MAGinator, we do know which subpopulation is the dominant one within each sample, even if its abundance estimate may be conflated with subdominant subpopulations. Additionally a section has been added to the introduction to clear up this concern: “A limitation of MAGinator is that the level of resolution of the output reflects the quality of the input contigs and MAGs.”

I want to once again highlight that I am receptive to any clarifications from the authors, and I sincerely hope that my understanding of how MAGinator works might be incorrect.

Best,
Reviewer

Reviewer #2 (Remarks to the Author):

The authors describe MAGinator, a new method for profiling of metagenomic data sets, able to identify strain-level variation and abundance estimates. The article is well-written, but could be a little more concise in parts, e.g. the introduction of HMO-associated microbes, some command line options (could be moved to the supplement) and the strain tracking results. The presented results are impressive and a significant improvement over the state-of-the-art for subspecies level profiling.

Reply 3) Thank you for the feedback, we have changed the manuscript according to the feedback. We have rewritten the part concerning HMO's in the introduction as well as specified the strain-tracking tracking results. We introduced the optional parameters as Supplementary table 2.

The software is available on GitHub and indeed installed easily. Unfortunately, I was not able to reproduce the results because of run time restrictions on the server available to me. Some information about the estimated running time should be included.

Reply 4) We've taken note of your feedback and have included instructions in the README to help adjust the parameters for better compatibility with one's setup. One suggestion is to consider modifying the --max_cores and --max_mem settings. While this adjustment may impact the speed of MAGinator as the parallelization of jobs will be limited, it might enhance its performance on your system. Finally, we have provided an expected runtime for the test dataset based on a run on our own server. Thank you for bringing this to our attention, and we hope these adjustments prove helpful and will allow MAGinator to run on your setup.

There are a few additional points which should be addressed:

Page 4: Has MAGinator been tested with other binners than VAMB? How does binning performance/MAG quality influence performance of the pipeline?

Reply 5) We chose VAMB as our binner of choice as it has demonstrated advantages in accuracy over other binners, but more importantly because MAG clustering is built in. I.e. MAGs are grouped across samples into MAG clusters. This information is essential for MAGinator and was taken from the default output of VAMB. Testing other binners for use with MAGinator requires that we first benchmark different methods and thresholds for MAG clustering (e.g. dREP <https://www.nature.com/articles/ismej2017126>), and we think this is a subject for separate study. We thank the reviewer for noting this area for potential improvement and are currently working on this in a follow-up analysis.

Page 5: The sentence "if they are consistently detected in similar abundance across samples, it suggests that they are single-copy" is not clear to me; a gene consistently occurring multiple times would also occur in similar abundances across samples?

Reply 6) We appreciate the reviewer's insightful observation regarding the clarity of the sentence. The reviewer is correct in pointing out that a gene occurring multiple times would also lead to similar abundances across samples. However, we model the data with the negative binomial model using the number of reads mapping to the MAG cluster as input. This means we know the expected prevalence of signature genes given the abundance of the MAG. This step was specifically employed to capture single-copy genes, thus sorting away multi-copy genes. If a gene should be multi-copy this would be penalised by these two metrics. However, should a signature gene still be multicopy, it is worthy to

note that this would result in a systematic, study-wide error consistently underestimating abundances of that MAG cluster across all samples, thus having a minimal or no impact on downstream association analysis against sample metadata. Similar errors are even more pronounced with amplicon sequencing metagenomics, as 16S copy numbers vary tremendously between bacterial taxa, and this is further compounded by primer biases. However, such errors have not prevented the detection of biologically meaningful signals with amplicon sequencing, as errors tend to be study-wide, and cancel each other out across samples.

Page 6: Why are no long indels expected between strains in different samples? The acquisition of e.g. amr genes in response to external influences seems possible.

Reply 7) Your insight is appreciated. It's entirely possible for long indels to emerge between strains in different samples. If a gene happens to be linked to an indel within a specific subpopulation, the selection criteria would circumvent the selection of these as part of the signature gene set. I.e. such genes would not be selected as part of the signature gene set, and other genes would be selected instead.

The reasoning for the values why a sample is included into the phylogeny should be explained somewhere

Reply 8) We've made efforts to clarify the rationale behind sample inclusion in the phylogenetic trees within the methods section. Additionally a description of the various user-adjustable settings guiding the sample inclusion in the phylogeny can now be found in Supplementary Table 2.

Page 7: Presented are the results on the CAMI 2 strain madness data set, but not on the other CAMI 2 data sets with fewer samples. Since the other data sets also consist of many samples, I am wondering whether there is an effect of the number of samples on the results by MAGinator. Is there a "minimum" number of samples for which MAGinator produces good results?

Reply 9) Indeed, MAGinator's performance is notably influenced by the number of samples included in the input data. Furthermore the results are also highly influenced by the complexity of the microbial environment. The more complex the environment, the more samples are needed to carry out the analysis. We wanted to test MAGinator on the Earth Microbiome Project dataset (<https://earthmicrobiome.org>), however due to its shallow sequencing and very high complexity, despite having 818 samples, from VAMB only 6 MAG clusters was produced containing >200,000kb bases (run with default settings), which would be unusable as input for MAGinator.

Larger sample sizes generally yield better results, as information from cross-sample analysis is used for both VAMB and MAGinator. However different parameters can also be changed in order to tailor MAGinator to the input data such as --binsize.

Figure 2/Suppl. Table 2/3: DUDes_cami1 reports a L1-error in Figure 2B, but in Suppl. Table 2 it has 0% completeness and 0% purity, how is this possible?

Reply 10) OPAL defines the L1 norm error as the absolute difference between the true and predicted abundances ranging from 0 (perfect reconstruction) to 2 (totally incorrect reconstruction). DUDes_cami1 predicts several strains, however none of them are true positives. This leads to an L1 norm error of 2. The purity and completeness are defined as percentages, with 0 as the worst and 1 as the perfect prediction of taxa. As no taxa are correctly predicted this leaves the completeness and purity of 0% for DUDes_cami1.

We have added the range of the L1 norm error to the manuscript.

Is there a reason for the - relatively - poor performance in terms of L1-error of MAGinator for the high phylogenetic ranks?

Reply 11) We used the gold standard and corresponding profiles of the benchmarked tools, which

they provide annotated using CAMITAX. To benchmark we used CAMITAX on the MAG cluster identified with MAGinator. In order to be able to compare the taxonomy at all phylogenetic levels the whole lineage must be found for each taxonomic ID. However for certain entries the NCBI-lineage (using taxidlineage.dmp, downloaded May 2023) did not match the lineage found in the gold standard, e.g. taxid: 1235803 (species-level), which had the following differences:

```
NCBI:      2|976|200643|171549|2005525|375288|2649774|1235803
CAMITAX:  2|976|200643|171549|171551|375288| |1235803| |
```

According to the most recent NCBI-database (updated 2024-01-03) the lineage is still the same as in May.

Despite the taxonomy agreeing on species-level a discrepancy is found at family-level. These differences in the databases likely result in the large error observed for the higher taxonomic ranks.

For the new gene synteny clusters, 80% of modules having 80% agreement is a nice result, though there are still >7,000 clusters for which there is no corresponding KEGG module, so some way to estimate or reduce the number of false positives would help.

Reply 12) *This is indeed a valid point. We believe many of the false negatives are due to the KEGG database not being comprehensive enough. I.e. there are still numerous pathways especially in bacteria for which a KEGG module has not yet been defined. This should improve over time.*

As for the false positives, we believe that these are mainly a result of our synteny algorithm lacking operon awareness. As we use genomic adjacency as the criterion for synteny, genes at either end of an operon that border unrelated operons could get lumped together when the compared host genomes are too closely related, thus producing inconsistent KEGG annotations. Using operonic co-occurrence instead of genomic adjacency would resolve this problem. However, accurate operon prediction is outside the scope of MAGinator.

That said, the user has quite some control over trading off sensitivity for specificity by setting custom thresholds for gene similarity and co-occurrence frequency. E.g. by setting "--synteny_mcl_inflation" to say 2 instead of 5, or "--clustering_min_seq_id" to 90% instead of 95%, synteny clusters would need to span increasingly unrelated genomes, and survive more genomic neighbourhood shuffling, thereby cancelling out spurious cooccurrences. The default thresholds were set conservatively leaving users free to experiment.

Reviewer #1 (Remarks to the Author):

I appreciate the responses from the authors. However, I have to say that my concern remains resolved.

From my point of view, the title of the manuscript might be misleading and MAGinator provided quantifications for the closely related "subspecies" in the same community might be meaningless. This is due to the fact that the sequences of closely related "subspecies" are most likely to be coupled together and hard to differentiate with the technologies (including sequencing, assembly, and binning) widely used today. In other words, if the sequences in a MAG are a mixture of sequences from multiple closely related "subspecies", it becomes meaningless to quantify these "subspecies" as we can't even distinguish them, we can only estimate the total abundance of these subspecies based on the MAG at hand.

I acknowledge that the authors have highlighted that the resolution and granularity of MAGinator's performance depend on the quality of the contig assembly and MAGs. However, I am concerned that MAGs we obtained with currently available technologies are rarely good enough to meeting MAGinator's assumption, that is, the sequences within a MAG originate from a single population, even in the presence of closely related populations within the same community.

I acknowledge all the hard work the authors put into preparing this manuscript. If the authors do not agree with my comments, I would suggest seeking a third viewer on their work.

Reviewer #2 (Remarks to the Author):

The questions raised were answered sufficiently by the authors, but some of the answers also should be part of the main manuscript or at least the supplement. In particular a comment about the high number of required samples (reply 9) and information about the tuning of sensitivity/specificity of the gene synteny clusters (reply 12) should be added to the manuscript.

REVIEWER COMMENTS

Reviewer #1 (Remarks to the Author):

I appreciate the responses from the authors. However, I have to say that my concern remains resolved.

From my point of view, the title of the manuscript might be misleading and MAGinator provided quantifications for the closely related “subspecies” in the same community might be meaningless. This is due to the fact that the sequences of closely related “subspecies” are most likely to be coupled together and hard to differentiate with the technologies (including sequencing, assembly, and binning) widely used today. In other words, if the sequences in a MAG are a mixture of sequences from multiple closely related “subspecies”, it becomes meaningless to quantify these “subspecies” as we can’t even distinguish them, we can only estimate the total abundance of these subspecies based on the MAG at hand.

I acknowledge that the authors have highlighted that the resolution and granularity of MAGinator’s performance depend on the quality of the contig assembly and MAGs.

However, I am concerned that MAGs we obtained with currently available technologies are rarely good enough to meeting MAGinator’s assumption, that is, the sequences within a MAG originate from a single population, even in the presence of closely related populations within the same community.

I acknowledge all the hard work the authors put into preparing this manuscript. If the authors do not agree with my comments, I would suggest seeking a third viewer on their work.

Reply 1) We appreciate these very valid reservations and agree with the reviewer that MAGs can be mixtures of subspecies in a single community which cannot always be distinguished. Our reason for putting strain/subspecies in the title was originally based on two points. 1) The observation that the upstream binner, Vamb in our case, could sometimes distinguish between closely related subspecies as separate MAG clusters, as exemplified by the case of B. longum longum/infantis presented in the paper. However, we fully acknowledge that this will not necessarily be possible in every dataset. And most MAG clusters indeed seem to represent species-level resolution. However, 2) when considering variation between samples, MAGinator provides phylogenies/trees for each MAG cluster based on read mappings to its signature genes. This variation is thus by definition within species.

Therefore, we have changed the title to better reflect this crucial distinction:

“MAGinator enables accurate profiling of de novo MAGs with strain-level phylogenies”

Furthermore, we have clarified this in the results section.

We see how we did not succeed in adequately describing the MAG cluster phylogeny feature of MAGinator. In response to the reviewer’s concern, we have revised the manuscript to provide a more nuanced explanation of MAGinator’s capabilities and limitations and highlighted how the user can identify scenarios where sequences within a MAG may originate from multiple closely related populations.

When a single sample contains a mixture of subspecies, we agree that the MAG phylogeny would be unreliable and the abundance calculation would encompass this mixture. Such cases can be flagged using the statistics provided for each MAG cluster and sample. We have improved the description on how the user can visualise this information. We have also added a subfigure C to supplementary figure 7 as an example:

The figure displays the phylogeny of a MAG cluster based on SNVs at the read level. We have now added a heatmap of the median frequency of bases of the signature gene alignment with mixed alleles. A high number would indicate that several subpopulations exist within the sample. For most samples this number is close to 0 (green), indicating that such samples are not conflated at the subspecies or strain level.

Overall, for the 716 MAG clusters in the COPSAC 2010 data set, 387 MAG clusters had no samples containing mixed alleles. 329 MAG clusters did harbour samples with mixed alleles, and within these particular MAG clusters an average of 38% of the samples had mixed alleles. In summary, across all MAG clusters, 4,154 of 20,765 leaf tips/samples had mixed alleles. This information is readily available from the output files.

We hope this explanation clarifies our intent, resolves the main disagreements, and ultimately that the paper presents the features of MAGinator to the reader in a more precise way. If it does not, we would very much like to refine the solution to ensure these points are fully met.

Reviewer #2 (Remarks to the Author):

The questions raised were answered sufficiently by the authors, but some of the answers also should be part of the main manuscript or at least the supplement. In particular a comment about the high number of required samples (reply 9) and information about the tuning of sensitivity/specificity of the gene synteny clusters (reply 12) should be added to the manuscript.

Reply 2) Your insight is appreciated. In the manuscript we have elaborated the relevant sections regarding number of samples required (reply 9) and regarding the construction of the synteny clusters (reply 12). Additionally we have added a comment relating to (reply 5) regarding uses of alternative binning tools and the necessity of MAG clustering before it can be used by MAGinator. Additionally we have elaborated the section, further explaining how the signature genes of one dataset can be used for analysis of another - smaller or shallow sequenced - data set or how the samples from multiple studies can be pooled in order to obtain reliable MAG clusters if they originate from similar environments.

Reviewer #3 (Remarks to the Author):

I feel the authors have mostly addressed the comments from reviewers 1 and 2. The change in the title more accurately reflects the manuscript results and I appreciate the analysis of SNV frequencies.

However I worry that users may still misinterpret the output of MAGinator when the MAG represents a mixture of strains.

For this reason, MAGinator should automatically flag MAGs that represent strain mixtures. Currently, (I believe) MAGinator reports the allele frequency and leaves it up to the user to interpret the result. It would be helpful if the authors recommended and/or applied a cutoff for determining a pure population MAG or a strain mixture MAG.

The authors evaluated the extent of strain mixing on MAGs from human gut metagenomes from the COPSAC 2010 data set. However, strain diversity is substantially higher in non-human-associated environments.

To address these challenges, the authors should profile strain mixtures on MAGs from other environments and determine the % of MAGs from each biome that pass the strain purity threshold.

REVIEWER COMMENTS

Reviewer #3 (Remarks to the Author):

I feel the authors have mostly addressed the comments from reviewers 1 and 2. The change in the title more accurately reflects the manuscript results and I appreciate the analysis of SNV frequencies.

However I worry that users may still misinterpret the output of MAGinator when the MAG represents a mixture of strains.

For this reason, MAGinator should automatically flag MAGs that represent strain mixtures. Currently, (I believe) MAGinator reports the allele frequency and leaves it up to the user to interpret the result. It would be helpful if the authors recommended and/or applied a cutoff for determining a pure population MAG or a strain mixture MAG.

Reply 1) We appreciate this concern and agree with the reviewer that we could assist the user with regards to interpreting these results. To address this we have added the generation of a table providing strain-mixture information for each MAG cluster for each sample. The provided table contain 3 different values:

0 - not present in the sample

1 - pure population, e.g. the average median allele frequency of the signature gene alignment is not above a certain threshold

2 - mixed strain population, e.g. allele frequency above the given threshold.

The threshold for the allele frequency determining a strain mixture has been set as a user-adjustable parameter, "--af_cutoff". The default value has been conservatively set to 0, indicating that if even one signature gene has a median allele frequency over 0, then the concerned MAG cluster will be flagged as mixed within the sample.

From the table, the user has a clear overview of each MAG cluster and the proportion of mixed/pure occurrences within their data.

The authors evaluated the extent of strain mixing on MAGs from human gut metagenomes from the COPSAC 2010 data set. However, strain diversity is substantially higher in non-human-associated environments.

To address these challenges, the authors should profile strain mixtures on MAGs from other environments and determine the % of MAGs from each biome that pass the strain purity threshold.

Reply 2) We have taken your advice into consideration and have expanded our analysis to include MAGs from honeybee and Tara Oceans datasets. As expected our findings show higher strain diversity across these environments. The honeybee microbiome exhibits lower species diversity but higher strain diversity compared to both the human microbiome and the ocean samples. The ocean samples contain slightly higher species diversity and higher strain diversity compared to the human gut metagenome. We have included Supplementary Table 5 to provide a summary of the results across the different environments.

Reviewer #3 (Remarks to the Author):

My concerns have been addressed.

I have two final comments regarding the result on lines 599-609, in which the authors briefly discuss the strain diversity results.

(1) Please provide users guidance on interpreting this information when analyzing results from the tool? Do we expect MAGs to be less reliable when strain diversity is high? For example, was the MAG completeness and N50 higher when strain diversity was low?

(2) The strain diversity metric has the potential to be used as another useful quality filter (similar to completeness and contamination). However, as it is currently implemented it would discard 86% of MAGs from marine samples. I'd suggest classifying samples into strain diversity tiers: None, Low, Medium, High, in order to give users more granularity into the data.

Reviewer #3 (Remarks to the Author):

My concerns have been addressed.

I have two final comments regarding the result on lines 599-609, in which the authors briefly discuss the strain diversity results.

(1) Please provide users guidance on interpreting this information when analyzing results from the tool? Do we expect MAGs to be less reliable when strain diversity is high? For example, was the MAG completeness and N50 higher when strain diversity was low?

Reply 1: The allele frequency-matrix provides information of strain-mixtures for each MAG cluster for each sample. We have analyzed the results of the COPSAC2010-cohort. We have created the quality-metrics for the MAGs with CheckM2 (v.1.0.2 run in predict-mode with default settings) - completeness, contamination and N50. For the MAGs we have calculated the average quality for the two types of populations.

	Average Completeness	Average Contamination	Average N50
1 - pure population	80	1.46	81,387
2 - mixed strain population	77.6	1.56	88,088

From the results we see that there is no clear difference between the MAGs containing pure vs mixed strain populations regarding MAG quality. To us, this indicates that assembly and binning quality is not necessarily influenced by this fine-grained level of diversity that is picked up by the MAGinator as strain mixtures, but we cannot make strong conclusions at this point.

(2) The strain diversity metric has the potential to be used as another useful quality filter (similar to completeness and contamination). However, as it is currently implemented it would discard 86% of MAGs from marine samples. I'd suggest classifying samples into strain diversity tiers: None, Low, Medium, High, in order to give users more granularity into the data.

Reply 2: We appreciate the reviewers suggestion to use the metric as an additional quality filter and agree that this could be useful depending on the user's research question. This functionality is already supported in our implementation through the user-adjustable parameter for allele frequencies, "--af_cutoff". While the default value is conservatively set to 0 - indicating that if even one signature gene has a median allele frequency over 0, the corresponding MAG cluster is flagged as mixed - users have the flexibility to adjust this threshold to better suit their specific research question and provide the extra granularity. From the results in Reply 1 this metric seems to capture something complementary to MAG quality. But if the user is interested in only pure populations, this can be used for sorting the data.